# A Review of Skin-Wearable Sensors for Non-Invasive Health Monitoring Applications

**DOI:** 10.3390/s23073673

**Published:** 2023-03-31

**Authors:** Pengsu Mao, Haoran Li, Zhibin Yu

**Affiliations:** 1Department of Industrial and Manufacturing Engineering, FAMU-FSU College of Engineering, Florida State University, Tallahassee, FL 32310, USA; 2High-Performance Materials Institute, Florida State University, Tallahassee, FL 32310, USA

**Keywords:** wearable sensors, health monitoring, non-invasive, stretchable, big data

## Abstract

The early detection of fatal diseases is crucial for medical diagnostics and treatment, both of which benefit the individual and society. Portable devices, such as thermometers and blood pressure monitors, and large instruments, such as computed tomography (CT) and X-ray scanners, have already been implemented to collect health-related information. However, collecting health information using conventional medical equipment at home or in a hospital can be inefficient and can potentially affect the timeliness of treatment. Therefore, on-time vital signal collection via healthcare monitoring has received increasing attention. As the largest organ of the human body, skin delivers significant signals reflecting our health condition; thus, receiving vital signals directly from the skin offers the opportunity for accessible and versatile non-invasive monitoring. In particular, emerging flexible and stretchable electronics demonstrate the capability of skin-like devices for on-time and continuous long-term health monitoring. Compared to traditional electronic devices, this type of device has better mechanical properties, such as skin conformal attachment, and maintains compatible detectability. This review divides the health information that can be obtained from skin using the sensor aspect’s input energy forms into five categories: thermoelectrical signals, neural electrical signals, photoelectrical signals, electrochemical signals, and mechanical pressure signals. We then summarize current skin-wearable health monitoring devices and provide outlooks on future development.

## 1. Introduction

Health-related issues are among the most vital topics affecting the continuation of humanity. Since the last century, modern medical systems and techniques have developed rapidly, prolonging lives by fighting against illness. While routine physical checkups and post-morbid consultations are two major approaches used to track many health problems, the danger of sudden illness still threatens our lives. Therefore, opportune medical diagnosis and treatment have gradually taken center stage. On-time continuous health monitoring offers us an opportunity to identify diseases in an early stage and minimize risks. Although it is easy to implement continuous monitoring in hospitals, limited medical resources and expensive medical expenses are prohibitive, especially for areas with unequal resources. To detect diseases in an early stage and provide greater accessibility to the public, one of the success solution paths is focusing on non-invasive health monitoring for on-time detection, data collection, and initial diagnosis. Our skin, which works as the interface between the deep tissues and the external environment, is an effective and suitable medium for non-invasive health monitoring, since it can reflect important signals related to our health status. Due to the accessibility of skin and the natural sensors we are equipped with, the earliest record of exploring physiological signals from skin can be tracked back to 400 BC, when people first detected that body temperature can be an index of illness [1]. The natural sensors of our body allow us to perceive temperature, pressure, taste, odor, light, etc. Advances in solid-state electronics over the last century have expanded horizons in sensing, leading to today’s sensors outputting electrical signals that are collected and processed almost entirely by computers. Such technology has been clinically utilized for health monitoring and has been continuously improved in terms of high sensitivity and safety; however, most applications are limited to clinics due to being bulky [2], expensive [3], and immobile [4], and requiring easy-drying electrolyte gels [5,6], all of which ultimately lead to difficulties in long-term monitoring [7]. In addition, the electronic components need to be attached to the skin due to inflexible rigid contact [8,9,10,11,12]. The vulnerable, non-conformal adhesive design leads to low measurement sensitivity and significant measurement error, especially during movement [13]. To overcome these limitations, a new generation of skin-wearable devices has emerged. This review includes studies and learnings on flexible and stretchable non-invasive skin-wearable electronics and classifies them according to their input energy form, i.e., thermoelectrical signals, neural electrical signals, photoelectrical signals, mechanical signals, and electrochemical signals.

## 2. Systematic Review of Skin-Wearable Health Monitoring Strategies

The epidermis and dermis bilayer of skin can be treated as a three-dimensional network of collagen fibers that straighten during stretching parallel to the applied load direction [14,15]. This property results in a high dependence between human skin and applied force orientation. The stretchability of the skin depends on whether the fibers are oriented. The fibers that cross Langer’s lines are less stretchable than fibers that are parallel to the lines [16]. The elastic deform strain of the skin can range from 20 to 30% with elastic modules ranging from 1.11 kpa to 57 Mpa depending on age, skin location, and testing methods [17,18,19,20,21,22,23,24]. Such behavior can be treated as the soft substrate for skin-wearable devices, and similar mechanical properties across electronic devices are required to achieve conformal contact with skin and to help patients tolerate the deformation during normal body stretching.

In addition to the substrates, the mechanical properties of sensing elements are also significant. This problem can be solved by three approaches: material selection, substrate pre-strain, and device structure. Solution-processed electronics and dispersible conducting/semiconducting nanomaterials are applied in the two major approaches to enhance electronic flexibility from a material aspect. The solution-processed sensors can be deposited onto the large-area soft substrates that are solvent-compatible with the material [25]. The manufacturing techniques, including spin coating, inkjet printing, spray coating, and doctor blade printing, are particularly useful for organic materials such as thermoelectric and optoelectronic materials [25,26,27]. However, the flexibility/stretchability of the devices is limited by both substrate and sensing elements. The stretchability can be greatly enhanced by embedding dispersible conducting/semiconducting nanomaterials, including nanoparticles [5], nanowires [28], nanotubes [29,30], and graphene into an elastomer matrix [30]. The preferred growth direction of the nanowires is difficult to control due to the spontaneous network formation of the 3D structure [31]. In addition, elastomeric materials have been used to embed/support rigid sensing components to achieve device stretchability. In the recent study, a bioadhesive hydrogel elastomer couplant was used as the stretchable substrate for rigid ultrasound probss [32]. The device can be attached to plenty of locations on our body for 48 h to continuously monitor acoustic signals from blood flow, the lungs, and the heart. The second approach is to apply a pre-strain to the stretchable substrate that uses buckles perpendicular to the axis, which impart elastic properties to the non-stretchable organic electronics [33]. The result shows that when applying 80% pre-strain on the stretchable substrate before applying the solution-processed organic electronics, the organic electronic device is able to maintain the electric property with restretching strain <80% [34]. Although this approach provides stretchability to non-stretchable organic electronics, the device cannot be stretched beyond the value and direction of pre-strain. Additionally, the devices cannot return to their original length after fabrication and release of pre-strain. The third strategy is using the filamentary serpentine (FS) or fractal geometry for sensing elements and electrodes [31]. The FS exploits the high-order iterative self-similar fractal patterns to design the conductive traces with strain orientations. This kind of curve can fill any arbitrary shape in a small area, maximize skin-electronic contact area, and effectively reduce the impedance. The sensing part is generally made by sandwiching ultrathin gold/chromium (Cr/Au) traces between polyimide(PI) layers due to their high resistance to atmospheric corrosion, the low resistivity of the Cr/Au combination, and the adhesion improvement of Cr [35]. The fractal traces can be further mounted on skin by van der Waal forces or pressure sensitive adhesives (PSA). More practically, FS traces can be designed in different layouts with adjustable resistance-based responses for multifunctional sensing, including body temperature [4], biopotential [36], and auxiliary strain sensing [31,37]. Furthermore, the combination of FS design with a pre-strained substrate can significantly improve the stretchability of the device [38,39].

Human skin is a highly complex biological material that insulates and protects internal organs from the external environment. It can be described as a multilayered structure comprising the epidermis, dermis, and hypodermis [17]. Epidermis (average thickness of 40–50 um) is the outermost layer of skin. The outermost layer of the epidermis is the stratum corneum, which is the main reason that the lifespan of skin-wearable devices is limited to 2 weeks due to the exfoliation of old cells [8]. The dermis layer, which is beneath the epidermis accounts for 90% of the thickness of the skin (1.5 to 4 mm) [40]. The dermis regulates body temperature and supplies oxygen and nutrients to the skin via blood vessels [40]. Sweat glands located throughout the body adjust body temperature by bringing water to the skin surface where it evaporates through pores. This process can produce up to 2 L of sweat per hour, providing a wealth of information about the metabolism [40]. Lymph vessels in the dermis contain infection-fighting cells from the immune system and play an important role in wound healing by destroying infecting or invading organisms [41,42]. Moreover, nerve endings that exist in the dermis layer transmit sensations, such as pressure, pain, and temperature to the brain for interpretation [40]. The hypodermis layer is the innermost layer of the skin, through which the blood vessels, nerves, lymph vessels, and hair follicles also pass. Since the light transmittance of human skin varies with wavelength, systolic–diastolic signals, and blood oxygen saturation can be detected from the blood vessels. High-sensitivity pressure sensors can record similar arterial systolic–diastolic signals due to the softness of the skin. Another signal that can be obtained from the skin surface is biopotential, which comes from the muscle layer beneath our skin [43]. Therefore, methods that detect electrophysiological signals, such as electroencephalography (EEG), electromyography (EMG), and electrocardiogram (ECG), can be used to monitor brain, heart, and muscle activity, respectively.

Skin-accessible health information can be obtained by an electronic sensor, which generally refers to devices that convert detected input signals into electrical outputs [44]. Based on the physical understanding of the signal domains, in 1989, Göpel et al. introduced the classification of transduction principles, which provides a method to visualize sensing principles [45]. While modern sensors can be more complex due to the involvement of multiple sensing principles and can be classified according to applications, specification, etc., this review selects the primary input energy form for classification in terms of material properties and selection. 

So far, the signals that can be transmitted and processed from the skin are divided into five types of sensors: thermal (of temperature), electrical (of neural electricity), radiant (of photoelectricity), mechanical (of pressure) (all of which are found in Table 1), and electrochemical (of sweat).

## 3. Electrical Signals Related to Neural Activity 

### 3.1. Neural Electrical Signal Monitoring Mechanism

Biological fluid activities that present electrolytes generate potentials. The potentials can be recorded with a direct electrical contact to skin or through capacitive coupling. The exploration of these activities and the molecular and cellular processes that control their signaling is called electrophysiology, a branch of neuroscience [72]. There are three main types of electrophysical signals that can be obtained non-invasively from skin: electrocardiology (ECG) for heart activity, electroencephalography (EEG) for brain activity, and electromyography (EMG) for muscle movement. 

The ECG signal is an electrical signal that travels through the heart by a series of depolarizations and repolarizations in cardiac cells [73]. This electrical signal is generated by an impulse from the depolarization of the atria (upper chamber of the heart) and presents as the P-wave of ECG. Then, the electrical current travels to the lower chamber of the heart, causing the depolarization of the ventricular cardiac muscle and presenting as QRS-complex. The P and QRS complexes have differences in amplitude due to differences in the muscle mass of the atria and ventricles. Repolarization then occurs in the opposite direction as the relaxation of the ventricular cardiac muscle and presents as T-wave (Figure 1). This signal includes an AC potential range from 0.5 to 5.0 mV (1 mV peak-to-peak), a DC potential component up to ±300 mV caused by impedance between the skin and the device, and a bandwidth range of 0.5–100 Hz [74]. Although ECG signals can be simply obtained by 2 electrodes, the clinic-standardized ECG uses 10 electrodes placed throughout the body based on Einthoven’s triangle and records the signals from 12 different directions [75,76]. Viewing the heart from unique angles in each direction provides the opportunity to localize lesioned areas [77]. The abnormality in ECG relates to cardiovascular diseases, such as arrhythmia [78], ischemia [79], hypertrophy [80], and aneurysm [81]. For conventional ECG recordings, the potential changes in 12 different directions in 10 electrodes are measured [76]. One common placement of the electrodes is based on Einthoven’s triangle, which is an imaginary triangle drawn around the volume of the heart [75]. Each apex of the triangle represents where the body’s composition, with respect to the heart, connects electrically with the limbs. The twelve leads are made up of three bipolar and nine monopolar leads [76]. There are three main leads responsible for measuring the electrical potential difference between arms and legs [74]. The three bipolar leads measure the electrical potential between the right and left arm (lead I), the right arm and the left foot (lead II), and between the left arm and the left foot (lead III). In all ECG lead measurements, the electrode connected to the right leg is considered the ground node. An ECG signal will be acquired using a biopotential amplifier and then displayed using instrumentation software, where a gain control will be created to adjust its amplitude. To minimize the size of the portable device, the smaller devices often rely on only two electrodes to deliver a single lead [82]. It is worth mentioning that cardiac activity includes muscle activity, a galvanic potential of arteriole vs. venule blood flows components, and a galvanic potential dynamic in the blood peripheral vessels.

EEG detects the postsynaptic potential of pyramidal cells in the epithelial cortex of the scalp [84,85]. The normal activities of a living individual can be divided into five EEG patterns, namely, Delta (deep sleep), Theta (under emotional pressure), Alpha (relaxed state), Beta (alert state), and Gamma (perceptual activity) [86]. With increasing activity, the frequency of EEG increases (from 0.3 to 50 Hz) and the amplitude decreases (from 200 to 5 μV) [86]. In normal brain activity, cells are activated asynchronously and deliver small potentials to the scalp. Neurological disorders, such as epileptic seizure, manifest as sudden synchronous and repetitive discharge by a group of simultaneously activated cells and produce high EEG amplitudes [85]. Therefore, EEG is commonly used to analyze neurological and psychiatric disturbances [87], such as brain disorders [88], epilepsy [89], brain tumors [90], brain injuries [91], and early stages of Parkinson’s disease [92]. An EEG system consists of electrodes, amplifiers, filters, and a recording unit [86]. Electrodes are usually placed on conductive gel, following the international 10–20-based system. The electrical signal received from the electrodes needs adequate amplification followed by the removal of noise, using an amplifier and a filter, respectively. An analog-to-digital converter (ADC) is employed to convert the signal into digital form before feeding it to a computer for analysis and storage. 

EMG is used to detect muscle movements and the nerve cells that control movements. Electrical signals of muscle movements are generated by the motor cortex of the brain, are transmitted to the spinal cord, and are finally sent to the relevant muscles via motor neurons [93]. During this process, upper motor neurons transmit information to lower motor neurons, which innervate muscle movement by releasing calcium ions and causing muscle tension [93,94]. The depolarization involved in the process provides an electrical current difference that can be detected by small needle electrodes on the skin. The measured electric signal can be less than 50 μV and as high as 30 mV, depending on different muscles [95]. Myopathic muscles result in more complex signals than normal muscle movement because they are generally smaller and need to be activated at a higher firing rate. In contrast, the signals of neurogenic disorders are less complex due to the fewer number of them needed to be activated at equivalent muscle activations [96]. Thus, diseases such as myopathy [97], neurogenic muscle wasting and weakness [98], abnormalities [99], radicular pathology [100], and motoneuron disease can be detected and diagnosed by EMG [101]. During an EMG test, small needles (electrodes) are inserted through the skin into muscle. The electrical activity picked up by the electrodes is then displayed on an oscilloscope. An audio-amplifier is used to hear the activity. EMG measures the electrical activity of muscle during slight contraction and forceful contraction. EMG clinical examinations have, thus, been extended for the detection of other diseases, including myopathy, neurogenic muscle wasting and weakness, abnormalities, radicular pathology, motor neuron disease, etc [100]. Furthermore, the EMG signals from muscle contractions of the biceps or triceps are commonly used for robotic control prosthetics, and machine-assisted living [8]. 

### 3.2. Neural Electrical Signal for Skin-Wearable Devices

Current clinical applications of ECG, EEG, and EMG use conductive gels (Ag/AgCl), which create less impedance between the skin and the electronic devices. The drying properties of conductive gels prevent long-term monitoring, which some sudden illnesses, such as epileptic seizure that does not continue to trigger epileptic discharges, require to document abnormalities [102]. An alternative is to use dry electrodes; however, dry electrodes suffer from high impedance and low signal-to-noise ratios (SNR), and create uncomfortable skin–device contact [103]. Since dry electrodes create weak signals and are susceptible to noise, it is imperative to explore the new electrophysical skin-wearable sensors with excellent mechanical properties for long-term attachment purposes. As summarized in Table 2, until now, proposed skin-wearable sensors for biopotential are utilizing two main strategies for conformal skin contact: van der Waals forces, adhesion, and pressure sensitive adhesion (PSA). 

Van der Waals forces include attraction and repulsion between surfaces, atoms, molecules, and intermolecular forces. Despite their short range (0.2–40 nm), van der Waals forces play an important role in microscale and nanoscale [106]. The devices that attach on the skin via van der Waals forces are capable of obtaining electrophysical signals on various areas of the body even under strain from different device structures and materials. Such dry bonding typically requires the participation of a low modulus and/or an ultra-thin gauge elastomer matrix. For example, carbon nanotube (CNT)-mixed polydimethylsiloxane (PDMS) was used to substitute the commercial Ag/AgCl gel for dry contact with skin [105]. Later research introduced ethoxylated polyethylenimine (PEIE) in PDMS/CNT composites to reduce impedance between the electrode and the skin [46]. The nature-inspired gecko tow pad structure further enhances the adhesion by replicating the columnar microstructure and fabricating it with silicone elastomer for strong and reversible adhesion between surfaces. The commercial conductive gel can be then substituted by this structure combined with commercial ECG electrodes [47]. The device is capable of continuously acquiring data for 48 h on flat or curved surfaces (chest and wrist) [47]. Silver (Ag) microparticles have been mixed with PDMS to form pillar structures, which exhibited an impedance of 50 ${k\Omega \;\mathrm{cm}}^{2}$ at 10 Hz and a readable high-fidelity ECG during movements [5]. Adding hybrid nanofiller (1D carbon CNT and 2D graphene nanopowder at a ratio of 1:9) to PDMS exhibits superior stretchability of over 100% and high conductivity, with a resistivity of less than 100 Ω cm, which is comparable to commercial ECG applications (Figure 2a–c) [30]. Van der Waals forces between the skin and the device can be achieved using low modulus silicone elastomers and sacrificial polymers. The sensing element consists of Cr/Au FS traces that are sandwiched between spin-coated polyimide and is accomplished by photolithography and dry etching. This method was first proposed in 2011 when researchers achieved conformal attachment of PI-Cr/Au-PI traces using low-modulus silicone (Figure 2d–f) [36]. The [polyvinyl alcohol (PVA)] is used as a temporary support because the water-soluble nanofibers allow it to stick on the skin via van der Waals forces. The device is stretchable with a repeatable strain of approximately 30%. By incorporating FS traces, the device is capable of multifunctional operation not only for ECG, EEG, and EMG but also for body temperature, LED and photodetectors, and wireless power.

The van der Waals forces approach can also be utilized for EMG signals and simulation. A sensor combined with an actuator employs a PI-Au-PI trace-like design on a water-soluble PVA substrate (Figure 2g–i) [37]. Here, the FS traces made of the same material with different geometries are used to detect EMG, temperature, and strain. The resistance-based results are adjustable to compensate for transverse sensitivity and eliminate the effects between different working elements. The device expands the scope of applications through the participation of actuators that receive electrical input and execute the movement of the robotic arm by placing two devices on the triceps and biceps, respectively. Inducing muscle contractions offers an opportunity for prosthetic control through perceptual feedback and stimulation. Additionally, van der Waals force attachments are eligible for complex textures in small areas, such as auricula, which require extremely bendable electronics. The attached tripolar concentric ring layout sensor with a PVA water-sacrificial layer can be laminated on the auricle to not only obtain long-term EEG signals but also track letter spelling by steady-state visually evoked potentials (SSVEP) [48]. Since each visual target flickers at a unique frequency, the algorithms based on canonical correlation analysis (CCA) are able to classify the subject’s desired character for test spelling. 

Another strategy for skin-to-device contact is the application of pressure sensitive adhesives (PSA), which are the viscous substrate that achieve conformal contact by electrostatic forces. Similar to van der Waals forces in essence, electrostatic interactions are attractive or repulsive forces between two surfaces caused by the opposite or the same charges [107]. The difference between them is that electrostatic force has a larger value and a longer dominant distance of a few micrometers [108,109]. It is worth mentioning that even when SPA tape is able to hold two surfaces together by electrostatic forces, van der Waals forces are still involved when two subjects are close enough. If the applied pressure is inadequate, the bonding faults, such as bubbles or detachment, may exist and affect adhesion [110]. Adhesion and flexibility contribute to PSA for wearable devices to form conformal contact even under large strain. In 2014, the skin-wearable ECG patch was made of mixing CNT with adhesive polydimethylsiloxane (aPDMS) as the adhesion layer for Au/Ti/polyimide FS to form a triangle structure with three leads and to provide ECG signals from three directions [49]. An adhesive force of 1.1${\;N\;\mathrm{cm}}^{-2}$ ensures the firm skin attachment with an impedance of 241 kΩ at 40 Hz. The device obtained comparable results to the commercial Ag/AgCl electrodes when attached to the subject’s chest. PSA layers enable large-area multi-channel devices for biopotential detection (Figure 2j–l) [50]. In this study, adhesive silicone was used as the interface between the skin and the device. The micropores in the adhesive silicone provide a highly breathable interface, which is the result of micropores poly (methyl methacrylate) (PMMA) that is embedded before curing and removed after curing by acetone. The device can measure reinnervated muscle movements simultaneously in eight channels when mounted on the amputated upper limb. Clear signals for ECG and EEG are obtained by using standard digital filtering techniques. Conformal electronics can further monitor these signals during functional magnetic resonance imaging (fMRI) and provide additional information on physical and psychological states [111]. Another way to enhance the adhesion force is to use an auxiliary bilayer consisting of a ferromagnetic layer and adhesive silicone with conductive FS traces [51]. The obtained device with an SNR of 8.2–13 is suitable for clinical applications, such as ECG, EEG, and EMG. In addition, conductive materials can also be mixed with structural materials to form the PSA layer directly. Using poly(ethylenedioxythiophene):poly(styrenesulfonate) (PEDOT:PSS) mixed with waterborne polyurethane (WPU) and D-sorbitol, the stretchable electronic with an adhesive force of 0.4 ${N\;\mathrm{cm}}^{-1}$ and a low impedance of 82 ${k\Omega \;\mathrm{cm}}^{2}$ at 10 Hz was formed [2]. The device can be attached to the skin to detect the clear signals of ECG, EMG, and EEG. 

## 4. Thermoelectrical Signal Measurement 

### 4.1. Thermoelectrical Signal Monitoring Mechanism 

Body temperature, a consequential by-product of heat energy metabolism, has been widely used as an index of illness since 400 BC [1,112,113]. The core energy of the body must be emitted to the surrounding environment to avoid overheat [114]. Heat transfer properties of the skin reveal changes relevant to human physiology, such as skin state, activity, and thermoregulation [57]. The heat production rate and the heat lost rate are maintained in a balance through radiation, convection, conduction, and evaporation, resulting in body temperature falling within a fairly narrow range from 36 to 38 ℃ [112,114]. Lost control of body temperature can lead to impairments of physiological function, lost consciousness, and/or death [112].

Body thermal detection has been widely used in the medical monitoring of inflammatory response, a defense mechanism of the immune system that evolved in higher organisms to protect them from infection and injury [115]. The purpose of inflammation is to locate and eliminate harmful substances and damaged tissue components, allowing the infected area of the wound to begin to heal [116]. The constriction of capillaries draws blood away from the infected area, causing engorgement of the capillary network, which increases blood flow and temperature in the wound healing area. Therefore, thermography, a process in which a thermal imager captures the images of infrared radiation emitted from the object, enables the localization and monitoring of the inflammation area and wound processes. In addition, thermal detection can prevent the patient from suffering a stroke or heart attack if such inflammation occurs in carotid arteries [117]. The evidence also suggests that the lesioned area of breast cancer release more heat than the surrounding areas, which could be caused by a number of possible explanations, including angiogenesis, nitric oxide, inflammation, and estrogen [118,119]. Thermography was used as an early preclinical diagnosis of breast cancerous lesions in 1956, and was approved by the FDA as a breast risk assessment tool in 1982 [120,121]. Finally, high skin temperature, especially during exercise, is associated with dehydration, which leads the cardiovascular and thermoregulatory systems under stress [112]. High skin temperature also results in plasma loss, which is caused by decreases in cutaneous capillary resistance due to profound cutaneous vasodilation [122].

### 4.2. Thermoelectrical Signal for Skin-Wearable Devices

Conventional skin temperature detection either uses simple single-point contact measurements or sophisticated infrared digital cameras for spatial imaging. Point contact methods are cost-effective and convenient for diseases such as systemic fever; however, it is hard to detect diseases that only occur in specific areas of the body, such as inflammation and breast cancer. Infrared spatial thermography offers an accurate solution for imaging, but it is expensive, incapable of continuous long-term monitoring, and requires the immobilization of patients [4]. The skin-wearable devices effectively solve this dilemma using three types of thermoelectric sensors: a resistance temperature sensor, a thermocouple, and a diode thermal sensor.

Resistance temperature sensors are made of metal and, therefore, have a positive temperature coefficient and lower sensitivity [123]. The temperature depends on resistance: (1)$$R=R_{0}\left(1+AT+BT^{2}\right)\;$$
where $R_{0}$ is the resistance at the reference temperature (usually 0 ℃), and *A* and *B* are the coefficients that depend on the different metals applied. The metals used most often are platinum (Pt), nickel (Ni), and copper (Cu). Cu has been built into a resistance thermal sensor to monitor wound healing by utilizing a 3-um-thick fractal layout designed in six sensing elements for an epidermis electronics system (EES) (Figure 3a–c) [52]. Having six sensors provides accessibility for spatial records of temperature and thermal conductivity. Polyimide layers and Cu traces minimize the bending strains. The encapsulated traces are transferred to a silicone membrane to realize a waterproof device. The results show that the electrical resistance change is capable of temperature measurement with a precision of ~50 mK in lab and ~200 mK in clinical testing. The device was laminated on a cutaneous wound from day 1 to day 30 after surgery to monitor the temperature change in the skin surface of the forearm. The inflammatory phase of wound healing can be detected when the thermal conductivity changes on day 3. Similar resistance-based metal filamentary serpentine designs also use Pt and Au as the sensing array for temperature sensing, with performances comparable to that of an infrared camera [4,36]. Human skin exhibits large variations in thermal properties. A Au coil geometry device has been reported to measure temperature up to 6 mm beneath the skin surface by finite element analysis (FEA) [53]. NiO has been used for body temperature sensing due to its relatively large negative temperature coefficient (NTC), which exhibits a simple structure and high temperature sensitivity [54]. In the study, the NiO metallic nanoparticles coated layer was irradiated with a laser to fabricate Ni-NiO-Ni structures with tens-of-micrometer-wide NiO channels. The final device presents a response time of 50 ms, which could be attributed to the 25 μm-thin PET substrate. The respiration temperature during exercise can be detected when the device is attached to the front of the nasal cavity. In addition, a metal and metal oxide, PEDOT:PSS, has also been used for temperature sensing by applying pre-strain to PDMS substrates [55].

A thermocouple measurement is based on the interaction between thermal energy and electrical energy and is associated with the Seebeck effect [123]. The Seebeck effect is the phenomenon of current flowing around a loop and/or electric potential across an open loop connecting two different types of metals with a temperature difference [124]. The performance of a thermocouple is described as the metals’ figure of merit:(2)$$ZT=S^{2}\sigma T/\kappa \;$$
where S is the Seebeck, σ is electrical conductivity, T is absolute temperature, and κ is thermal conductivity [125]. High electrical conductivity and the Seebeck coefficient, along with low thermal conductivity, contribute to the performance of the thermoelectric materials [126]. A chromium/gold (Cr/Au) filamentary serpentine structure has been designed for the epidermal sensor that maps macrovascular and microvascular blood flow beneath a targeted area of skin (Figure 3d–f) [56]. The 3-mm-diameter device contains 1 central actuator (10-nm Cr/100-nm Au) and 2 rings from the sensors (10-nm Cr/100-nm Au) for spatial response. During the sensing process, the actuator provides a constant thermal source at the surface of the targeted vessel. The two sensors rings provide the result of the thermal distributions with a precision of 0.01 °C. An additional sensor outside the rings is served to detect and compensate the temperature changes of the surroundings. The sensor is supported, and it is adhered to the skin, by silicone substrates. The high gas permeability ($2{\;g\;h}^{-1}m^{-2}$) of the silicone base and the low thermal mass ($0.2-5.7{\;\mathrm{mJ}\;\mathrm{cm}}^{-2}K^{-1}$) of the complete device ensure the minimal perturbation of the skin temperature. The changes in blood flow were able to be recorded during hyperemic reaction induced occlusion and reperfusion on the skin surface of the subject. Furthermore, the correlation of skin temperature detected by the ultrathin Cr/Au thermocouple, which recorded vascularization, blood flow, stratum corneum thickness, and hydration level, has been reported [57]. 

A diode thermal sensor is attributed to the nonlinear asymmetric diffusion current in a p-n junction with two different materials that have different temperature-dependent thermal conductivities [127,128,129]. Adding an intrinsic layer (i-region) sandwiched between the p-layer and the n-layer constructs a p-i-n(PIN) diode, which has a wide intrinsic layer that provides low and constant capacitance, high breakdown voltage in reverse bias, and a variable attenuator [123]. Recent research has helped design a multiplexed sensor array based on PIN diodes formed by the patterned doping of Si nanomembranes for a skin-wearable thermal sensor (Figure 3g–i) [4]. The small thickness of the device results in a thermal mass of $7.2{\;\mathrm{mJ}\;\mathrm{cm}}^{-2}K^{-1}$ and a thermal inertia of $\sim 500{\;\mathrm{Ws}}^{1/2}m^{-2}K^{-1}$. The PIN system includes 64 sensors (each with a size of 100 um by 200 um) with 16 external connections. The resolution of the spatial image can, thus, be improved by the scalability of microfabrication. This sensing array displays the precision of ~8 mK in lab and ~14 mK in a hospital setting at 0.5 Hz.

## 5. Photoelectrical Signal Measurement

### 5.1. Photoelectrical Signal Monitoring Mechanism

The human body is light transparent, and this natural property has been used to detect pulsating arterial blood since World War II [130]. Light passing through our body is attenuated and absorbed by a pulsating component, such as blood passing through the arteries and arterioles, and a non-pulsating component, such as venous blood, bones, and other tissues [58]. The pulsating part is considered a DC component (noise signal), and the non-pulsating part is treated as an AC signal. The systolic is caused by a forward-going pressure wave that transmits from the left ventricle to the peripheral tissue, and the diastolic is the result of the pressure wave transmitting from the aorta to small arteries in the lower body [131]. The recorded systolic–diastolic cycle is shown in Figure 4a. This continuous change is recorded by a light source and a photodetector (PD) via PPG. According to the PPG signals in different areas of the body, the abnormality of blood vessels can be located [132]. Because PPG is caused by heart contraction and blood flow, the pulse transit time (PPT) exists between the systolic and diastolic peaks. It represents the propagation time of the pressure wave from the subclavian artery to the apparent reflection site and back to the subclavian artery [131,133]. The artery stiffness (SI), which reflects blood flow velocity, is proportional to the ratio between the subject’s height (h) and PPT [134]. The researchers observed that 98% of the patients with arteriosclerosis in the study had a diminution or disappearance of the dicrotic wave, while the health volunteers present ill-defined dicrotic peaks [135]. In addition, first and second derivative PPG signals are also related to the carotid artery [136], age [137], blood pressure [138], the risk estimation of coronary diseases [139], and the presence of atherosclerotic disorders [140]. Commonly used light sources for PPG are green light and red/near-infrared light. Green light is largely used in a commercial daily reflectance mode due to its great signal-to-noise ratio and ability to resist motion artifacts; however, it can be absorbed by melanin, resulting in an increase in a variation of accuracy [141,142]. Red/near-infrared light has been used for medical purposes because the human body does not absorb much red light and because it can go deeper into multiple tissue layers for biometric signals; however, red/near-infrared has a lower signal-to-noise ratio and susceptibility to motion artifacts [142]. It is worth mentioning that water begins to absorb light with a wavelength greater than 940 nm, and, therefore, the maximum wavelength of PPG is generally 940 nm [143,144]. 

PPG signals received at different wavelengths can be used to acquire ${\mathrm{SO}}_{2}$. In our body, hemoglobin plays an important role in carrying oxygen to our organs due to the limitation of oxygen dissolved in our blood (0.3 mL gaseous oxygen per 100 mL blood). The 100 mL blood contains up to 20.1 mL oxygen with the help of hemoglobin. One molecule of hemoglobin is only stable with four or zero oxygen molecules, and oxy-hemoglobin (${\mathrm{HbO}}_{2}$) and deoxy-hemoglobin (Hb) have different light absorptivity, especially with red (λ=660 nm) and near-infrared (λ=940 nm) light (Figure 4b) [145]. According to the Beer–Lambert Law, the absorption ratio of ${\mathrm{HbO}}_{2}$ and Hb can be used to calculate ${\mathrm{SO}}_{2}$ base on the light intensity attenuated by human tissue [58,146]; however, the result is limited by the tissue scattering (rather than a glass cuvette). In order to manufacture a commercial pulse oximeter, an empirical correction is required to overcome the limitation by analyzing the light absorption at two wavelengths from AC/DC volume [146,147]. 

Transmission and reflectance are two modes of accessing PPG and ${\mathrm{SO}}_{2}$. Transmission places the photodetector (PD) and light-emitting diodes (LEDs) on opposite sides of the tissue and detects the light signal that is transmitted through the vascular bed. In contrast, reflectance places the PD and LEDs on the same side of the body tissue and monitors the light signals that are reflected from the vascular bed [148]. Device placement in the transmission mode is limited because the device must be located on the body sites where light can be readily passed through. The most common sites for transmission the are fingertip and earlobe; however, these sites have limited blood perfusion and are susceptible to environmental extremes and daily activities [149]. Despite being under a static state reflectance presents less noise (due to less light scattering) [115], less fluctuation of the DC component (caused by a shorter light path) [13], and various possible placement sites throughout the body (even the wrists, forehead, ankle, and torso) [149,150] Reflectance is also more susceptible to motion artifacts and pressure disturbances than transmission [149]. During physical activity under reflectance, the pressure interface skin device acting on the sensing probe causes deformation and the distortion of the arterial geometry. In addition, measurement sites and source–detector separation distance may also be factors affecting the result obtained from reflectance [115,151].

### 5.2. Photoelectric Signal for Skin-Wearable Devices

Although commercial devices are now able to minimize the size of components, the issue of skin–device attachment caused by rigid electronics still exists. Non-conformal contact leads to noise, especially during body movement; however, applying too much pressure to skin through the strips leads to torsion of the vessels and inaccuracy of the results. In contrast, the soft PPG has the potential to solve the current problem even when the body is in a dynamic state.

One approach is to use solution-processed electronics, which have better mechanical properties compared to current commercial devices. An organic optoelectronic sensor is one option for a skin-wearable pulse oximeter; however, organic LED materials are not stable in air and show low efficiencies [58]. Therefore, an all-organic pulse oximeter with green light (λ = 532 nm) and red light (λ = 626 nm) has been proposed due to the different light absorbability of ${\mathrm{HbO}}_{2}$ and Hb at these wavelengths [58]. The result shows the ability of the device to detect PPG and ${\mathrm{SO}}_{2}$. Later, the green light (λ = 517 nm) and red light (λ = 609 nm) polymer LEDs were combined with an organic photodetector (poly(3-hexylthiophene) (P3HT): (6,6)-phenyl-C61_butyric acid methyl ester (PCBM)) to fabricate a soft pulse oximeter (Figure 4c–e) [59]. The passivation layers composed of organic parylene and inorganic SiON layers ensure that the LEDs can be operated in air up to 29 h. The soft and ultra-flexible pulse oximeter can be taped to a fingertip with clear pulsating curves for ${\mathrm{SO}}_{2}$ calibration. 

## 6. Mechanical Signal Measurement 

### 6.1. Mechanical Signal Monitoring Mechanism

The pulse and blood pressure from arteries have been detected extensively on the skin surface over the radial and brachial artery for a long history, and this has been attributed to the artery’s elasticity [152]. Although it is believed that the center aortic pulse wave is a better determination for clinical cardiovascular risk, direct measurement involves invasively inserting a catheter into the heart [153]. Currently, two non-invasive alternative methods have been applied to obtain aortic pressure by using mechanical pressure sensors in the field of arterial tonometry: the first is to record the pulse wave over the skin of the common carotid artery, which is considered to give a similar result as the carotid artery; and the second is to detect the pulse wave on the skin of the radial artery and apply the model to estimate the central pressure [154,155,156].

The schematic diagram of the obtained pulse wave is shown in Figure 5a [157], where systolic pressure (P1) is caused by left ventricular ejection, and renal reflection pressure (P2) and the iliac reflection pulse (P3) involve arterial blood propagating backward caused by the contraction of the left ventricle [158,159]. P2 represents the elasticity of the artery since the renal site is the junction of the aorta, where the diameter changes significantly. Therefore, the ratio between P2 and P1 is defined as the augmentation index (AI), which is the index of systemic arterial stiffness [157]. The stiffness of the arteries is related to the levels of collagen and elastin, which generally decrease with age and cause an increase in P2 [160,161]. Interestingly, in any age, women have a greater AI value than men, and yet women have lower cardiovascular risk than men [152,162]. Furthermore, the non-invasive pulse waves measured from the carotid and femoral arteries can be used to calculate pulse wave velocity (cfPWV). The cfPWV is measured by applanation to obtain arterial pulses at the carotid and femoral arteries, combined with the ECG reading as the time reference, and calculated using the subject’s height as a distance [163]. Since the carotid–femoral pathway includes the aorta and its first branches, which are primarily responsible for pathophysiological arterial stiffness, cfPWV is considered a “gold standard” for arterial stiffness [164,165,166]. The stiffer arteries present an increase in the cfPWV value with a cut-off value of 10 m/s [167,168]. Non-invasive cfPWV detection without physiological impact is of great significance for the recovery of patients after complex surgery (such as angiogenesis); however, current clinical detections apply planar and rigid formats, and are attached on skin by mechanical fixtures, such as straps and tapes [68]. Continuous monitoring with such devices is difficult, especially after patients are discharged from the hospital. 

Mechanical pressure sensors can also help with the assessment of skin diseases. The mechanical properties of human skin are mainly derived from the contributions of connective tissue, the dermis, subcutaneous tissue, and the epidermis [169]. The first works about the mechanical properties of human skin began with Dupuytren in 1834 and Langer in 1861, who mainly studied the anisotropy of the natural stress and strain fields of skin [170]. Changes in the mechanical properties of human skin reflect tissue modifications, which are caused by sex [170], aging [171], skin lesions [172], and the hydration of cosmetics [172]. Traditional measurements (such as suction, indentation, traction, torsion, and wave propagation) provide the results that are affected by external variables, such as pressure applied to skin, skin thickness, and detection area [36,170,173]. By contrast, soft skin-wearable pressure sensors can not only address these issues through conformal skin contact but also enable long-term monitoring in patients’ daily lives. Furthermore, recent studies have proposed skin-wearable ultrasound devices based on mechanical (acoustic) signal detection [32,71,174].

### 6.2. Mechanical Signal for Skin-Wearable Devices

Currently, the proposed skin-wearable pressure sensors mainly focus on three mechanisms: piezoresistivity, capacitance, and piezoelectricity (Table 3).

The transduction behavior of piezoresistivity, which transfers a mechanical displacement into an impedance change, is suitable for soft electronics due to its sensitivity to structure deformation caused by applied pressure [175]. Piezoresistive sensing has a simple read-out mechanism [176], and a fast response to structural deformation [177]. Adjusting the shape structure between the two sheets is key to improving the sensitivity of the piezoresistive sensor. However, piezoresistive sensors have limitations, such as temperature dependence [178], a powering supply requirement [179], and low sensitivity, especially with ultra-low pressure being applied [180]. Poly(dimethylsiloxane) (PDMS) was patterned into a micronized pyramid structure and coated with (poly(3,4-ethylenedioxythiophene–poly (styrenesulfonate) (PEDOT:PSS) and an aqueous polyurethane dispersion (PUD) composite polymer for pulse wave sensing (Figure 5b–d) [60]. The stretchable device with a sensitivity of $4.88{\;\mathrm{kPa}}^{-1}$ (0.37<ρ<5.9 kPa) can be attached on the wrist and can measure a pulse pressure of 170 Pa (or 1.3 mmHg) with the low operation voltage of 0.2 V. Another structure is the epidermis-inspired microstructure, which consists of Merkel disks that resemble abrasive paper topography [61]. In one study, PDMS was patterned on abrasive paper for a spinosum microstructure and coated with graphene oxide (GO). The device can detect physiological signals, such as pulse waves from the wrist, respiration states from the chest, and motion activities from the feet. The sensor was able to be attached to a volunteer’s arm for 2 days without discomfort or allergic reaction. Furthermore, a bioinspired beetles’ wing-locking structure has been used for piezoresistivity [62]. The supramolecular assembly of graphene oxide (GO) was first applied onto the hydrophobic polyurethane (PU) sponge and then obtained the conductive polyaniline nanohair (PANIH) arrays by in situ reduction. The device can withstand up to 80% deformation with a high sensitivity of $0.0021{\;\mathrm{kPa}}^{-1}$ (ρ<2.3 kPa). The device can also detect a low-frequency of 4–6 Hz, which provides the possibility of predicting early-stage Parkinson’s disease with imitated static tremors. Eutectic gallium–induium (EGaIn), the liquid metal alloys, can also be used to build a wearable piezoresistivity sensor by embedding it into microchannels made of PDMS [63]. The conductive material can be substituted by EGaIn due to its low toxicity and high electrical conductivity. In the study, EGaIn was formed into an equivalent Wheatstone bridge circuit. The device presents a high sensitivity of 0.0835 ${\mathrm{kPa}}^{-1}$ and 0.0834 ${\mathrm{kPa}}^{-1}$ for loading and unloading, respectively. The PDMS wristband of a 1.8 mm diameter pressure sensor was attached to the subjects to measure a clear output dynamic pulse wave. 

Capacitive sensors measure elastic deformation by detecting the changes in electrical capacitance. The capacitance (C) of a parallel plate capacitor is given as: (3)$$\;\;C=\varepsilon_{0}\;\varepsilon_{r}\;A/d\;\;\;$$
where $\varepsilon_{0}$ is the free space permittivity as the constant, $\varepsilon_{r}$ is the relative permittivity, A is the area, and d is the distance between electrodes [181]. The simplicity of the governing equation simplifies the device design and analysis [182]. Although capacitive sensors have high sensitivity, low power consumption, and resistance to temperature variation, they are susceptible to interference from external sources [181,183]. A pyramid structure has also been used for capacitive sensing in healthcare monitoring. The wearable electronic uses biocompatible materials that make it biodegradable. It is composed of Poly (glycerol sebacate) (PGS) to form the pyramid structure, polyhydroxybutyrate/polyhydropxyvalerate (PHB/PHV) as the substrate, magnesium (Mg) as the electrodes, and iron (Fe) as an adhesion layer for Mg (Figure 5e,f) [64]. The Young’s modulus of PGS is within range of 0.05–2 Mpa, which ensures that it can take the place of PDMS (Young’s modulus ~1 Mpa). The device presents the pressure sensitivity of $0.76{\;\mathrm{kPa}}^{-1}$ (ρ<2 kPa). The cfPWV was measured with the value of 7.5 m/s when applied the soft device onto the subject and used the ECG result as a time reference. MXene (Ti3C2Tx)/poly (vinylidene fluoride-trifluoroethylene) (PVDF-TrFE) composite nanofibrous scaffolds have been used as a dielectric layer between poly-(3,4-ethylenedioxythiophene) polystyrene sulfonate /polydimethylsiloxane (PEDOT:PSS) electrode layers to form a capacitive pressure sensor for pulse wave detection [65]. The sensitivity of the device can reach $0.51{\;\mathrm{kPa}}^{-1}$ with a minimum detection limit of 1.5 Pa. The device can be used to detect multiple pressure changes on the skin, such as a pressure pulse wave on the wrist, those caused by the initial stage Parkinson’s disease, muscle contraction and expansion, muscle vibrations during eye twitching, and voice recognition. Another alternative to PDMS is styrene-butadiene-styrene (SBS) due to low dielectric loss in the high frequency range [184]. In the study, the pyramid elastomer was patterned onto the spatial copper design, which was coated on polyimide [66]. The sensor achieved wireless operation by near-filed electromagnetic coupling between the resonant sensor and an external antenna. The design can capture the pulse wave with a pressure resolution of 0.3 mmHg and a time resolution of 90 ms and can even measure intracranial pressure when implanted in mice.

Piezoelectricity occurs in certain materials (such as crystal, ceramics, and piezoelectric or electroactive films) that accumulate electrical charge in response to an applied mechanical force, changing the distance between dipoles of the material [182,185]. The ability of a material to convert mechanical force to an electrical charge can be described as $d_{33}$, the piezoelectric coefficient:(4)d_33=σ/ρ
where σ is the charge density and ρ is the applied force [186]. Although piezoelectric sensors have high sensitivity, a rapid response, and the potential for self-power [187,188], they exhibit thermal drift over time [189]. Ferroelectric materials have been widely used in piezoelectric sensors. The researchers used a printed ferroelectric polymer, poly(vinylindene fluoride-co-trifluoro ethylene) [P(VDF-TrFE)]-based pressure detector to fabricate a flexible device with PEDOT:PSS electrodes and a layer of poly(ethylene naphthalate) (PEN) substrate [67]. The device also consists of organic thin-film transistors (OTFTs) in a Darlington connection to amplify the signal by a gain factor of 10. The design ensures a sensitivity below 10 kPa and a fast response time of ~0.1 s. It can monitor pulse waves and the cfPWV (9 m/s) of the subject. A muscle structure-inspired piezoelectric sensor has been made by dispersing polydopamine (PDA) into barium titanate/polyvinylidene fluoride (BTO/PVDF) nanofibers to form interfacial bonds through an active surface functional group [70]. The fabricated nanofiber layer was bilaterally laminated with aluminum foils and used electrodes and flexible PET as substrates to realize a wearable device with a sensitivity of $3.95{\;V\;N}^{-1}$. Lead zirconate titanate (PZT), the electric/piezoelectric material, has been arranged in an array for pulse wave and cfPWV sensing [68]. The detected signal can be amplified by a silicone nanomembranes (SiMNs) n-channel metal semiconductor field effect transistor (n-MOSFETs). The conductive metal serpentine traces and the silicone substrate ensure the device can be stretched up to ~ 30% while maintaining the sensitivity of ~ 0.005 Pa. Furthermore, PZT-based nanoribbons were constructed as ultrathin stretchable networks for in vivo viscoelasticity measurements of epidermis lesions (Figure 5g–i) [69]. Due to the water-soluble polyvinyl alcohol (PVA) and the filamentous serpentine traces on the silicone substrate, the device can be attached to the skin by van der Waals forces alone with adequate stretchability. The polyimide encapsulated capacitor components were fabricated by a layer of piezoelectric PZT interposed between a Ti/PT bottom layer and a Cr/Au top electrode. The conformal device consists of seven actuators and six sensors for detection. Sensing involves outputting sinusoidal voltage changes through a selected actuator to induce mechanical motions in the PZT, surrounding elastomer, and the underlying epidermis. The mechanical coupling of the skin and elastomer causes the deformation of adjacent PZT. The results obtained from 30 patient volunteers using the device showed that the moduli of skin lesions on the breast and legs were lower than that of healthy skin, while the moduli of the nose and forehead area were higher than that of healthy skin.

### 6.3. Acoustic Signals Measurement

An acoustic wave is a mechanical wave that propagates along or through a medium via interactions between particles. The human ear perceives an acoustic frequency from 0.02 to 20 kHz. Diagnostic ultrasound scanners operate in the frequency range of 2 to 18 MHz, which is much higher than the human ear perceives [190]. Higher frequencies have correspondingly shorter acoustic wavelengths, which produce higher-resolution images; however, shorter wavelengths prevent penetration into deeper tissues. Therefore, higher frequency ultrasound (20 to 50 MHz) benefits from high-resolution images with low tissue penetration depth and is currently widely applied in dermatological diagnosis and treatment [191]. 

An ultrasound pulse is produced by passing an electrical current through a piezoelectrical crystal element. The element converts the electrical energy into a mechanical ultrasound wave. The frequency of the wave is determined by the thickness of the crystal. The thicker crystal element produces lower frequencies while the thinner crystal element produces higher frequencies. When the wave propagates through materials of different densities in the body, a frequency shift occurs due to the different mechanical properties of the materials. The reflected echo returns to the transducer, which converts the shifted ultrasound wave into an electrical signal. Diagnostic sonography is an ultrasound-based medical imaging diagnostic technique that enables the visualization of muscles and internal organs, offering visuals of their size, structure, and potential pathological lesions. The continuous monitoring of deeper tissues and internal organs has clinical value; however, long-term monitoring is limited by unstable coupling to tissue surfaces due to the bulkiness and rigidity of the existing non-invasive ultrasound equipment [192]. In addition, since the ultrasonic wave cannot transmit in air, an ultrasonic coupling must be used and pressure must be applied to reduce impedance and facilitate the transmission of sound energy between the transducer and the test sites. Such requirements have resulted in ultrasonography being available on limited body sites, since pression over the carotid artery, jugular vein, and vagus nerve could cause suffocation [32]. Furthermore, sonographers also suffer from work-related musculoskeletal disorders due to the prolonged use of the transducer, repetitive motions, and static work postures [193,194].

One approach to long-term ultrasonic monitoring is to apply rigid ultrasound probes on a bioadhesive hydrogel elastomer coupling to form a bioadhesive ultrasound (BAUS) device (Figure 6a–c) [32]. The BAUS coupling consists of a hydrogel formed by chitosan-polyacrylamide interpenetrating polymer networks (10 wt%) and water (90 wt%). The hydrogel is encapsulated by an elastomer membrane of polyurethane. The hydrogel–elastomer hybrid is further coated by a thin bioadhesive layer, which can form physical bonds and electrostatic interactions with the skin. The BAUS probe consists of an array of piezoelectric elements, which enable a high density of elements and high-resolution images. With a center frequency of 3, 7, and 10 MHz, the electrical impedance of the probes at the center frequencies are 49, 36, and 43 ohms, respectively. The BAUS device can provide images of the blood vessels, stomach, lungs, diaphragm, and heart for over 48 h. 

The second approach is to fabricate a stretchable ultrasound probe to be worn on the skin. The stretchable ultrasound device can be built with metal traces, using electrodes and piezoelectric 1–3 composite as transducer materials [174]. In addition, the wearable ultrasound imager with high stretchability of up to approximately 110% has also been built to detect the critical cardiac features (Figure 6d-e) [71]. The device was built by utilizing liquid metal as the electrodes, triblock copolymer (styrene–ethylene–butylene–styrene (SEBS)) as the substrate, and 1–3 piezoelectric composite (PZT-5H) as the ultrasound transducer. The center resonant frequency of the device is 3 MHz and the penetration is greater than 16 cm. The device can also detect cardiac activities before, during, and after exercise. A deep learning model was applied to extract information, such as stroke volume, cardiac output, and ejection fraction, from sequential images. 

## 7. Electrochemical Signal Measurement

### 7.1. Electrochemical Signal Monitoring Mechanism

Human sweat is a biofluid that contains electrolytes and metabolic wastes that can reflect the condition of plasma [195]. It is produced by the secretory coils of eccrine glands and delivered to the surface of skin through dermal ducts (Figure 7a) [196]. The most accessible analytes in our sweat and the methods to detect them are shown in Table 4. The life-essential analytes depend on each other while having their own balance [197,198,199]. Concentrations beyond the range may cause certain diseases, which can be diagnosed early and treated via sweat detection. Among the analytes, the most abundant ion species are sodium (Na) and chloride (Cl), for which levels are elevated with an increase in sweat rate [199]. A concentration of ${\mathrm{Cl}}^{-}$ in excess of 60 mM indicates cystic fibrosis [200,201], an inherited chronic disease affecting the lungs and digestive system [202]. The depletion of ionized potassium ($K^{+}$) is associated with heat-illness [203]. Loss of ionized calcium (${\mathrm{Ca}}^{2+}$) in sweat accounts for up to 30% of daily body losses [204]. ${\mathrm{Ca}}^{2+}$ levels effect the structure of many organs and body systems. Low levels of ${\mathrm{Ca}}^{2+}$ can lead to disease, such as cirrhosis and renal failure [205,206], while high levels of ${\mathrm{Ca}}^{2+}$ can lead to hyperparathyroidism [207]. High ammonia levels can be used as markers of hepatic disorders, such as hepatitis or cirrhosis [208]. It is noticeable that ammonia levels in sweat are higher than in blood due to the high pH of sweat [209]. Changes in skin pH value can result in skin disorders such as dermatitis, atopic dermatitis, ichthyosis, acne vulgaris, and fungal infections [210]. Sweat lactate is a hallmark of pressure ischemia, manifested by insufficient oxidative metabolism and impaired tissue vitality [211]. So far, there is no evidence that shows that sweat lactate is related to blood lactate or skin pH value [212]. Similarly, a precise relationship between skin glucose and blood glucose requires further evidence of correlations, since at least three parts of skin can produce glucose (stratum corneum, outward migration of interstitial fluid, and sweat) [213]. Heavy metals, such as copper (Cu) and zinc (Zn), are also essential elements, but in limited amounts in the body [214]. Excessive Cu levels lead to Wilson diseases [215], while insufficient Zn levels affect the immune system by impacting wound healing and even DNA synthesis [214]. In addition to essential analytes, sweat secret toxic metals, such as cadmium (Cd), mercury (Hg), and lead (Pb), accumulate in our bodies for a lifetime without producing physiological benefits [216]. These toxic metals can enter our bodies in daily life. For example, Cd and Hg accumulate in seafoods and tobacco, and Cd, Hg, and Pb are all found in products such as electronics, batteries, and alloys [217].

**Table 4 sensors-23-03673-t004:** Analytes in sweat and their on-body detection methods.

Analyte		Concentration in Sweat [196]	Method	Substrate	Recognition Element	Refs
Sodium		10–100 mM	Potentiometry	Temporary tattoo	Sodium ionophore	[218]
				Adhesive Tape	Sodium ionophore	[219]
				PMMA	Sodium ionophore	[220]
				PET	Sodium ionophore	[200,221]
Chloride		10–100 mM		Polyester	Ag/AgCl	[201]
				PET	Ag/AgCl	[200]
Potassium		1–18.5 mM		PET	Potassium ionophore	[221]
Calcium		0.41–12.4 mM		PET	Calcium ionophore	[222]
Ammonia		0.1–1 mM		Temporary tattoo	Nonactin ionophore	[223]
Heavy metal	Pb	<100${\;\mu g\;L}^{-1}$	Square-wave stripping voltammetry	PET	Bismuth, gold	[224]
	Cd	<100${\;\mu g\;L}^{-1}$		PET	Bismuth	[224]
	Hg	<100${\;\mu g\;L}^{-1}$		PET	Gold	[224]
	Cu	100–1000${\;\mu g\;L}^{-1}$		PET	Gold	[224]
	Zn	100–1560${\;\mu g\;L}^{-1}$		PET	Bismuth	[224]
				Temporary tattoo	Bismuth	[225]
pH		3–8 (4–6.8)	Potentiometry	PET	Hydrogen ionophore	[222]
				Temporary tattoo	Poly(aniline)	[226]
				PMMA	Poly(aniline)	[220]
			Colorimetry	PMMA	Bromophenol green (BCG), bromophenol purple (BCP)	[227]
				PDMS	bromophenol purple (BCP)	[228]
Thermal properties		36.5–37.5 °C		PET	Thermochromic liquid crystals	[229]
Lactate		5–20 nM	Chronoamperometry	Temporary tattoo	Lactate oxidase	[230]
				Parylene	Lactate oxidase	[231]
				PMMA	Lactate oxidase	[220]
				PET	Lactate oxidase	[221]
Glucose		10–200 μM		PET	Glucose oxidase	[200,221]

### 7.2. Electrochemical Signal for Skin-Wearable Devices

Sweat is detected by a rigid collector that is attached to skin by applied pressure. The device absorbs sweat via capillary forces and tests the sweat via an off-body method. Sensitivity and the reliability of results are limited due to evaporation and chemical degradation during the process [209]. Therefore, an on-body device capable of continuous acquisition and sensing at the point of generation is demanded to overcome the limitations. 

One approach to on-body sensing is colorimetric detection, which has been used to detect skin thermal properties [229], pH value, and the levels of glucose, lactate, and chloride [228]. One study formed soft serpentine microchannels (3 mm in diameter) using PDMS to absorb sweat by capillary forces, and analyzed the relationship between color change and concentrations on site [228]. The device can measure total sweat loss over time and use ultraviolet (UV)-visible spectroscopy to quantify analyte concentration. It contains near field communication (NFC) electronics that enable wireless communication with smart devices. In addition, colorimetric exposure can be used to detect the thermal properties of skin using thermochromic liquid crystals [229].

Electrochemical detection provides an alternative approach due to the high sensitivity and rapid response of measuring currents or potentials on-site [209]. It includes potentiometry, chronoamperometry, and voltammetry. By applying potentiometry, iontophoretic electrode-induced sweat can be analyzed on-site immediately by highly sensitive sensors located on the same substrate as the relevant sensing targets (Figure 7b) [200]. Iontophoresis of sweat sensing is applied for transdermal drug delivery systems [232,233], which help ameliorate the symptoms of pain relief [234], chronic edema [235], and rheumatic condition [236]. Since the concentration of the target analyte is based on the potential difference between the sensing electrode and the reference electrodes, this method can only be used to detect charged species, such as ${\mathrm{Na}}^{+}$,${\;\mathrm{Cl}}^{-}$, $K^{+}$, ${\mathrm{Ca}}^{2+}$, and $H^{+}\left(\mathrm{pH}\right)$ [209]. Glucose, lactate, and ethanol can be detected by chronoamperometry, which relays on the specific enzymes to oxidize and generate a current proportional to their concentration [209]. Here is an example of the process for lactate [237]:(5)$$\mathrm{actate}+{\;O}_{2}\overset{\mathrm{Lactate}\;\mathrm{Oxidase}}{\to }\;\mathrm{Pyruvate}+{\;H}_{2}O_{2}$$
(6)$$H_{2}O_{2}\overset{\mathrm{Electrode}\;}{\to }\;2H^{+}+{\;O}_{2}+2e^{-}$$

In practice, one device can contain one or more sweat analysis methods. For example, one study integrated sensing elements on a PET substrate that simultaneously detected glucose, lactate, ${\mathrm{Na}}^{+}$, and $K^{+}$ via chronoamperometry and potentiometry (Figure 7c–e) [221]. The device also contained metallic traces to compensate for surrounding temperature. The circuit components, such as amplifiers, filters, and wireless transmission, are consolidated into a flexible printed circuit board (FPCB), and the device can be worn on the subject’s wrist. Furthermore, trace metals (${\mathrm{Cu}}^{2+}$, ${\mathrm{Zn}}^{2+}$, ${\mathrm{Cd}}^{2+}$, ${\mathrm{Pb}}^{2+}$, ${\mathrm{Hg}}^{+}$) can be detected by square wave anodic stripping voltammetry (SWASV), which uses gold and bismuth as electrodes. Researchers fabricated a soft sensing array on PET substrate with gold (Au) and bismuth (Bi) as the working electrodes (WE), silver as the reference electrode (RE), and Au as the counter electrode (CE) [224]. The integrated sensing part can be bent with curvature radii of 3.2 mm and can be attached to skin with a wristband. The result shows that an Au electrode is suitable for Pb, Cu, and Hg, while a Bi electrode is suitable for Zn, Cd, and Pb. 

Among secretions from skin other than sweat, the volatile organic compounds (VOCs) released from skin are also non-invasively detectable and can provide important clinical information [238]. Similarly to sweat, skin VOCs are a mixture of compounds that are generated from metabolic processes. It is known that breath is the largest contributor of VOCs in humans [239]. As the secondary source of VOCs, skin-emitted VOCs have a correlation with breath [240]. The VOCs from breath have been well studied, as the signals can be sensed by optical sensors, colorimetric sensors, surface acoustic wave sensors, piezoelectric sensors, metal oxide sensors, silicon nanowire sensors, and monolayer-coated metal nanoparticle sensors [241], while research from soft/stretchable skin-wearable VOC detection devices is limited. Moreover, VOCs testing cannot be applied for early diagnosis, and must be combined with other strategies to increase diagnostic accuracy [241]. Regardless, VOCs detection can be used as one of the important supplementary sources of health monitoring.

## 8. Future Challenges and Opportunities

Skin-wearable sensors offer the opportunity to read health information non-invasively through human skin. Soft skin-electronic contacts reduce impedance and improve signal-to-noise ratio. Tests of the reviewed sensors demonstrate the feasibility of their applications. Future devices can be improved from the following aspects.

Despite the fact that the lifespan of skin-wearable devices is constrained to 2 weeks, due to the exfoliation of the stratum corneum [8], few of the studies reviewed here achieved realistic monitoring up to 2 weeks (Table 5). In fact, most studies on conformal contact have not been tested for hours, and only few studies have demonstrated the two-week adhesion length. Wired power and communication are the main reasons for limiting long-term monitoring. Several studies have proposed solutions for wireless communication, such as stretchable solar cells [34], triboelectric mechanisms [242], bioelectrocatalytic reactions [243,244,245], soft solid batteries and capacitors for on-body power generation [246,247], and near field communication (NFC) [66,229]. Therefore, combining such technologies with skin-wearable sensors could potentially make products more valuable. Accelerated life testing can further examine the effective duration of a device’s lifespan.

Self-learning techniques can be further implemented on skin-wearable devices for immediate diagnosis and abnormality alerts. Current analyses of electrophysical signals are mainly conducted through visual assessment, which is considered time-consuming and as low-yield work. However, sudden illness, such as seizures, requires continuous signal monitoring and data reviewing for hours or even days [102]. Additionally, inconsistent conclusions will be drawn by the inspectors’ level of expertise. Combining convolutional neural networks similar to a multilayer perceptron with deep learning to automatically recognize and classify given information for the analysis of ECG, EEG, and EMG is urgently required these days [102,144,248]. This point-of-care detection and diagnostic technology has the potential to take skin-wearable sensors to a new era. Initiative actuators embedding provides the opportunity for disease treatment and rehabilitation. Current skin-wearable devices mainly focus on the passive monitoring of health messages. Early treatment is one of the improvements that broadens the application range of existing designs. So far, actuators have been integrated into wearable devices for EMG [37,50]. Through the self-learning process of practice and simulation, patients are able to drive prosthesis readily by inputting electric power.

Beyond health-related monitoring and quick diagnoses, safety assessments will drive more interest for wearable sensors/devices since they can provide strong safety-related crisis management systems for human being. The non-biocompatibility of skin-attached sensors can lead to many complications, including cell cytotoxicity, skin irritation, skin sensitization, and skin chronic inflammation, and can also lead to commercial product recalls due to reported dermatitis. The U.S. Food and Drug Administration (FDA) and the International Organization for Standardization (ISO) require that skin-contact devices be evaluated for cytotoxicity, sensitization, and irritation [249,250,251]. To validate skin-wearable devices for public use, skin–material compatibility cannot be ignored. 

In summary, skin-wearable sensors are interdisciplinary devices that involve medicine, materials, machinery, electrical hardware, software, etc. The development of skin-wearable sensors depends on the development of each discipline. Skin-wearable electronics can detect vital health information and have the potential to overcome the limitations of current rigid devices. Soft, conformal skin-electronic contacts provide opportunities for long-term continuous health monitoring with results comparable to commercial methods. In the foreseeable future, skin-wearable electronic devices should become a mainstream medical monitoring method because of improvements such as a wireless power supply and communication, self-learning diagnostic techniques, early treatment, and overall safety assessment. Through the combination of different technologies, wearable electronics can not only be used to monitor and diagnose diseases but also help disease treatment and rehabilitation. 

## Figures and Tables

**Figure 1 sensors-23-03673-f001:**
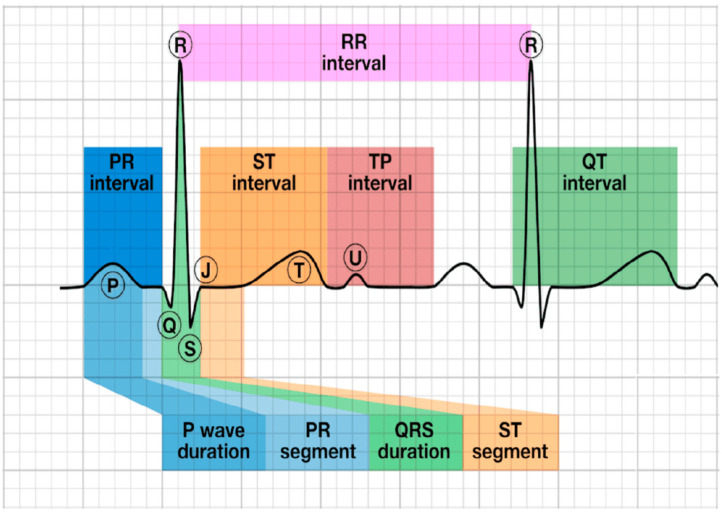
Schematic of ECG signal caused by depolarization and repolarization of the heart. (Reprinted/adapted from reference [83]).

**Figure 2 sensors-23-03673-f002:**
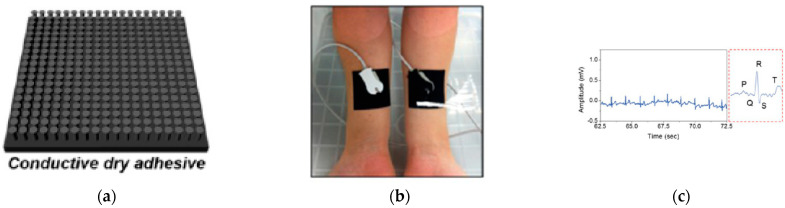
Images of skin-wearable electrophysical devices that are attached on skin based on van der Waals forces (**a**–**i**) and pressure sensitive adhesives (PSA) (**j**–**l**). (**a**) Schematic diagram of gecko-inspired micropatterns with nanocomposite mixed in elastomer matrix. (**b**) The device is capable of continuous monitoring on the forearm under water immersion conditions with (**c**) ECG signal. (**d**) Schematic diagram of multifunctional FS-based device. (**e**) The device on skin under compression. (**f**) Example of an ECG measurement from chest attachment. (**g**) Schematic diagram of the device design for EMG monitoring and simulation. (**h**) The devices on biceps and triceps for robotic arm control. (**i**) Separate EMG signals from the two devices during flexion and extension of the robot elbow angle. (**j**) Schematic diagram of layers of the large-area multi-channel flexible and stretchable device. (**k**) Conformal contact between the subject’s scale and the device with 68 electrodes for EEG detection. (**l**) The EEG and ECG signals obtained with standard digital filtering techniques. (Reprinted/adapted from reference [30]; Reprinted/adapted with permission from reference [36], Copyright 2011,Science; Reprinted/adapted with permission from reference [37], Copyright 2015, Advanced Materials; Reprinted/adapted with permission from reference [50], Copyright 2019, Nature Biomedical Engineering).

**Figure 3 sensors-23-03673-f003:**
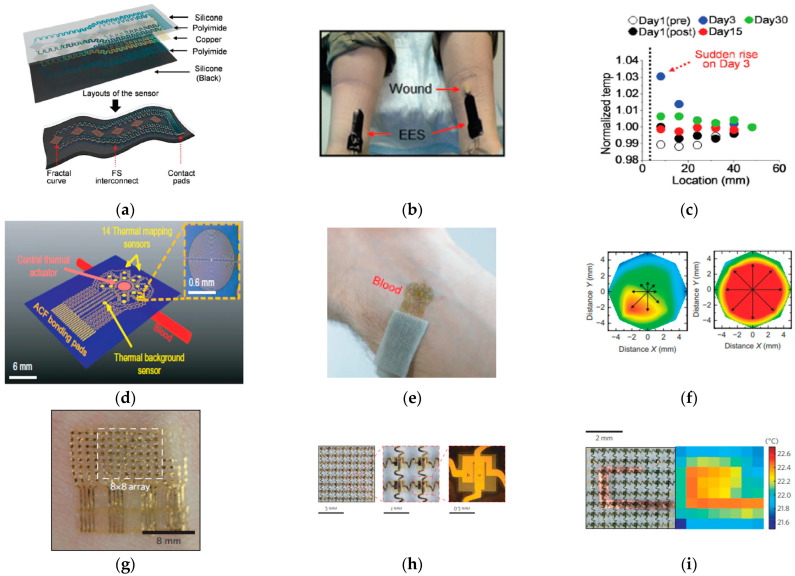
Images of skin-wearable thermoelectrical devices with spatial thermography. (**a**) Schematic of the multilayer structure of the resistance thermal sensor with six sensing elements of fractal curves and interconnecting filamentary traces. (**b**) Devices mounted on wound. (**c**) Temperature distribution recorded with 6 sensors spanning 45 mm in lateral direction starting near the wound site. (**d**) Schematic illustration of the thermocouple layout for an actuator and 14 thermal mapping sensors. (**e**) Placement of the sensor on human skin. (**f**) An example of the spatial thermography at peal flow (left) and occluded flow (right). (**g**) Optical image of an 8 by 8 Si nanomembrane diode sensor array mounted on the skin. (**h**) Optical images of the 8 by 8 Si nanomembrane diode sensor array (left) and magnified views of a single sensor (center and right). (**i**) Optical image of the device mounted on a heated Cu element (left) and measured distribution of temperature (right). (Reprinted/adapted with permission from reference [52], Copyright 2014, Advanced Healthcare Materials; Reprinted/adapted from reference [56]; Reprinted/adapted with permission from reference [4], Copyright 2013, Nature Materials).

**Figure 4 sensors-23-03673-f004:**
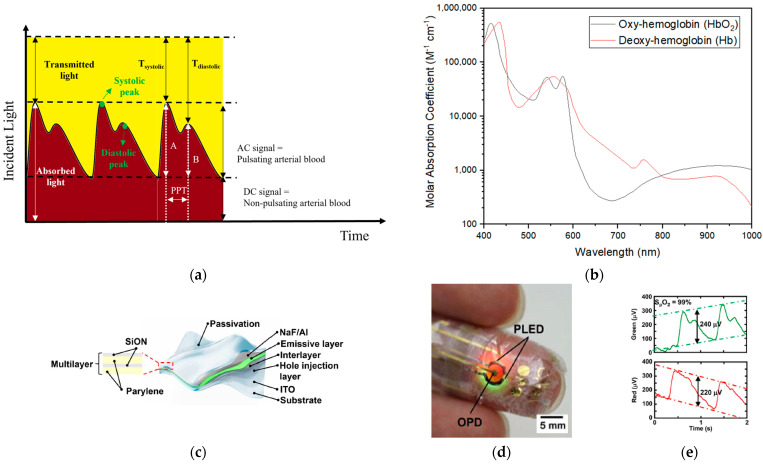
Images of photoelectrical signal measurement for skin-wearable devices. (**a**) The model for the pulse oximeter’s light transmission path. (**b**) Molar extinction coefficient for hemoglobin in water. (**c**) Schematic of polymer LED with passivation layer composed by organic parylene layers and inorganic SiON layers. (**d**) The soft pulse oximeter taped on fingertip. (**e**) The output green (top) and red (bottom) PPG signals with 99% oxygenation of blood. (Reprinted/adapted from reference [59]).

**Figure 5 sensors-23-03673-f005:**
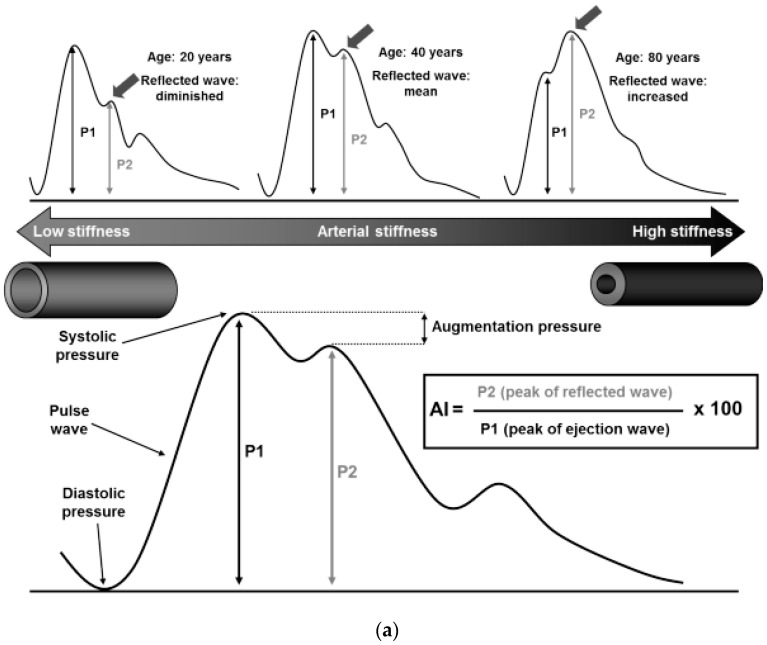
Images of skin-wearable mechanical pressure sensors to detect sweat analytes. (**a**) Schematic of the variables that determine the Augmentation Index (AI). (**b**) Circuit mode of piezoresistive sensor fabricated in PDMS pyramid structure and coated with PEDOT:PSS/PUD thin film. (**c**) The device is placed on the subject’s wrist (left) and the detected signal (right). (**d**) The detected important features of the pulse wave for health monitoring. (**e**) Schematic of the biodegradable and flexible capacitive pressure sensor. (**f**) Combined with ECG measurement, blood pulse waves detected by the capacitive pressure sensor can be used to measure pulse wave velocity. (**g**) Schematic of the piezoelectric device for skin biomechanics. (**h**) The conformal device mounted on the near nose region of the subject and (**i**) the collected date under actuation voltage and frequency of 5 V and 1 Hz, respectively. (Reprinted/adapted from reference [157]; Reprinted/adapted with permission from reference [60], Copyright 2014, Advanced Materials; Reprinted/adapted with permission from reference [64], Copyright 2015, Advanced Materials; Reprinted/adapted with permission from reference [69], Copyright 2015, Nature Materials).

**Figure 6 sensors-23-03673-f006:**
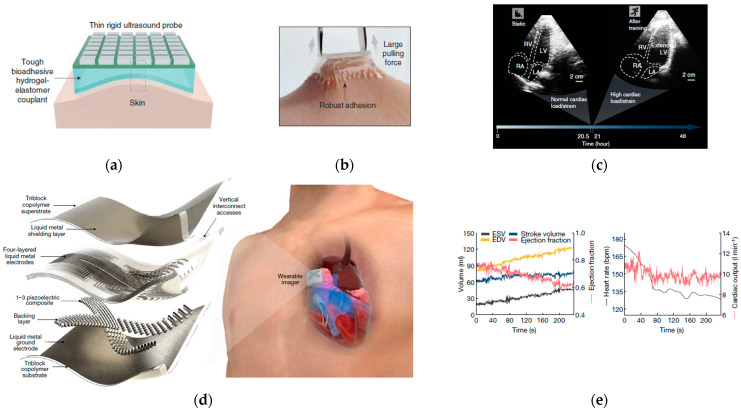
Images of skin-wearable acoustic signals. (**a**) Schematic of BAUS device, which consists of a thin and rigid ultrasound probe and can be attached to skin via a hydrogel–elastomer hybrid. (**b**) Skin-attached BAUS device under high pulling forces. (**c**) The BAUS imaging shows the dynamics of the four chambers of the heart. (**d**) Schematic diagram of the wearable cardiac ultrasound imager (left) and its working principle (right). (**e**) The extracted information of stroke volume, ejection fraction (left), and cardiac output (right) from consecutive images by a deep learning model. (Reprinted/adapted with permission from reference [32], Copyright 2022, Science; Reprinted/adapted from reference [71]).

**Figure 7 sensors-23-03673-f007:**
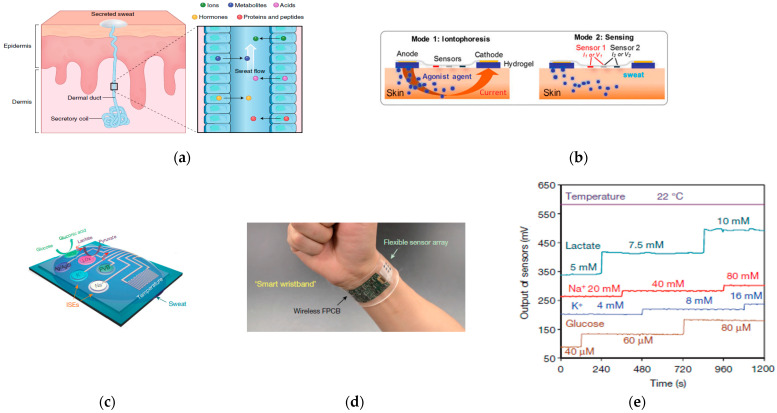
Images of skin-wearable electrochemical devices to detect sweat analytes. (**a**) Schematic of sweat gland and bioelectrolytes. (**b**) Illustrations of iontophoresis method for sweat sensing. (**c**) Schematic of the multisensing elements for sweat analytes. (**d**) Image of the device worn on subject’s wrist. (**e**) The device outputs multiple sense responses simultaneously. (Reprinted/adapted with permission from reference [196], Copyright 2016, Nature; Reprinted/adapted from reference [200]; Reprinted/adapted with permission from reference [221], Copyright 2016, Nature).

**Table 1 sensors-23-03673-t001:** Summary of skin-wearable devices.

Sensor Classification	Sensing Mechanism	Measurement Signal	Conductive Material	Signal	Measure Locations	Treatment	Substrate	Refs
Neural electrical sensor	Electrode	ECG	CNT	Voltage	Chest	Cardiology	PEIE/CNT/PDMS	[46]
		ECG	Commercial 3M electrode	Voltage	Chest and wrist	Cardiology, brain activity, muscle movement	PDMS	[47]
		ECG	Ag microparticles	Voltage	Chest, arm, scalp	Cardiology	PDMS	[5]
		ECG, EMG	Carbon nanofillers	Voltage	Wrist, stomach, ankle	Cardiology, muscle movement	PDMS	[30]
		ECG, EEG, EMG	Cr/Au FS	Voltage/frequency	Arm, neck, forehead, chest, leg	Cardiology, brain activity, muscle movement	Modified silicone (Smooth-on)/PVA	[36]
		EMG	Au FS	Voltage	Arm, prosthetic	Muscle movement, robotic arm control	Ecoflex (Smooth-on)/PVA	[37]
		EEG	Au FS	Voltage	Ear	Brain activity	silicone elastomer/PVA	[48]
		ECG	Au/Ti FS	Voltage	Neck, chest	Cardiology	Adhesive PDMS	[49]
		ECG, EEG, EMG	Cr/Au FS	Voltage	Back, arm, scalp	Cardiology, brain activity, prosthetic control	Silicone bylayer: Adhesive silicone (RT GEL 4642, Bluestar), Ecoflex (00-30 Smooth-on)	[50]
		ECG, EMG	Cr/Au FS with NdFeB	Voltage	Chest, arm, cheek	Cardiology, muscle movement	Adhesive Bluestar Silicones	[51]
		ECG, EEG, EMG	PEDOT:PSS	Voltage	Chest, arm, scalp	Cardiology, muscle movement, brain activity	PEDOT:PSS/ waterborne polyurethane (WPU) /D-sorbitol blend	[2]
Thermal sensor	Thermal resistance	Temperature	Cu	Resistance	Arm	Wound healing	Ecoflex	[52]
			Pt	Resistance	Skin	Body temperature	Modified silicone (Smooth-on)/PVA	[36]
			Au	Resistance	Arm	Body temperature	Ecoflex, (Smooth-on)/PVA	[37]
			Au	Resistance	Skin	Temperature beneath 6 mm of skin surface	Ecoflex	[53]
			Ni-NiO-Ni	Resistance	Facial surface	Respiration temperature	PET	[54]
			PEDOT:PSS	Resistance	Skin/hand	Body temperature	PDMS	[55]
	Thermocouple		Cr/Au FS	Voltage	Wrist	Blood flow	Ecoflex	[56]
			Cr/Au FS	Voltage	Cheek	Vascularization, blood flow, stratum corneum thickness, hydration	Ecoflex 00–30	[57]
	Diode thermal sensor		PIN diode sensor	Voltage	Skin, palm	Body temperature	Silicone elastomer/PVA	[4]
Photodetector	Photoelectricity	PPG, SO_2_	P3HT: PCBM	Voltage	Fingertip	Cardiology	Parylene	[58,59]
Mechanical sensor	Piezoresisticity	Blood pulse	PEDOT: PSS	Current	Wrist	Cardiology	PDMS	[60]
		Blood pulse	Graphene oxide	Resistance	Wrist, fingertip, chest	Cardiology, respiration states	PDMS	[61]
		Blood pulse, static tremor	Graphene oxide	Current	Wrist, fingertip	Cardiology, Early-stage Parkinson’s disease	Polyurethane sponge	[62]
		pulse rate	EGaIn	Voltage	Wrist	Cardiology	PDMS	[63]
	Capacitance	Pulse rate, PWV	Mg/Fe	Capacitance	Wrist, carotid artery, skin above femoral artery	Cardiology, arterial stiffness	PHB/PHV	[64]
		Pulse rate, muscle movement	PEDOT:PSS/ Mxene/P(VDF-TrFE)	Capacitance	Wrist, arm, dermal area of eye and throat	Cardiology, Early-stage Parkinson’s disease	PDMS	[65]
		Pulse rate, physiologic pressure	Cu	Capacitance	Wrist, Intracranial (of mice)	Cardiology, intracranial pressure	SBS	[66]
	Piezoelectricity	Pulse rate, PWV	P(VDF-TrFE) /PEDOT:PSS	Voltage	Wrist, neck	Cardiology, arterial stiffness	PEN	[67]
		Pulse rate, PWV	PZT	Current	Wrist, neck,	Cardiology, arterial stiffness	Ecoflex 00-30	[68]
		Skin modulus	PZT	Voltage	pathologies of skin regions	Dermatology	Ecoflex 00-30	[69]
		Pulse rate	PDA/BTO/PCVF	Voltage	Wrist, dermal area of throat	Cardiology	PET	[70]
		Ultrasound images	Commercial ultrasound prob	Voltage	Neck, chest	Images of lung, diaphragm, heart, and stomach	Bioadhesive hydrogel elastomer couplant	[32]
		Ultrasound images	PZT-5H	Voltage	Chest	Images of heart	SEBS	[71]

**Table 2 sensors-23-03673-t002:** Summary of electrophysiology sensors.

Skin-Contact Strategy	Skin-Contact Method/Material	Electrode Material	Impedance	Treatment	Refs
Van der Waals forces	Hydrogel electrode	Ag/AgCl	20 kΩ (50 Hz)~75 kΩ (100 Hz)	ECG	[104]
	CNT/PDMS	CNT	~100 kΩ (100 Hz)	ECG	[105]
	CNT/PEIE/PDMS	CNT	~145 kΩ (100 Hz)	ECG	[46]
	PDMS /in gecko-inspired micropillar structure	Commercial medical 3M electrode	N/A	ECG	[47]
	Ag particles in PDMS/ in micropillar structure	Ag	50 kΩ (10 hz)	ECG, EEG, EMG	[5]
	hybrid nanofiller in PDMS/ in gecko-inspired micropillar structure	1-D CNT and 2-D graphene nanopower	N/A	ECG	[30]
	Silicone elastomer/PVA	Cr/Au FS	N/A	ECG, EEG, EMG	[36]
	Ecoflex (Smooth-on) /PVA	Au FS	N/A	EMG	[37]
	Silicone elastomer/PVA	Au FS	N/A	EEG	[48]
Pressure sensitive adhesive layer	CNT/Adhesive PDMS	Au/Ti/polyimide FS	241 kΩ (40 Hz)	ECG	[49]
	Silicone adhesive	Cr/Au FS	30 kΩ (30 Hz)	ECG, EEG, EEG	[50]
	Adhesive Bluestar Silicones / ferromagnetic dipole	Cr/Au FS	<100 kΩ (10 kHz)	ECG, EEG, EMG	[51]
	PEDOT:PSS/ waterborne polyurethane (WPU) /D-sorbitol	PEDOT:PSS	82 ${k\Omega \;\mathrm{cm}}^{2}$ at 10 Hz	ECG, EEG, EMG	[2]

**Table 3 sensors-23-03673-t003:** Summary of pressure sensors and their properties.

Pressure Sensor Classification	Sensitivity	Response Time	Stretchability	Operating Voltage	Healthcare Monitoring	Refs
Piezoresistivity	$4.88{\;\mathrm{kPa}}^{-1}$ (0.37<ρ<5.9 kPa)	0.2 s	50%	0.2 v	Wrist blood wave	[60]
	$25.1{\;\mathrm{kPa}}^{-1}$ (ρ<2.6 kPa)	80–120 ms	N/A	N/A	physiological signals, voice, and motion activities	[61]
	$0.0021{\;\mathrm{kPa}}^{-1}$ (ρ<2.3 kPa); $0.0044{\;\mathrm{kPa}}^{-1}-$ $0.0109{\;\mathrm{kPa}}^{-1}$ (2.3 kPa<ρ<80 kPa)	22 ms	80%	N/A	Early-stage Parkinson’s disease (tremor of 4–6 Hz)	[62]
	$0.0834{\;\mathrm{kPa}}^{-1}$– $0.0835{\;\mathrm{kPa}}^{-1}$ (ρ<0.8 MPa)	90 ms	Stretching > 200% strain without failure	30 mV	Cardiology	[63]
Capacitance	$0.76\pm 0.14{\;\mathrm{kPa}}^{-1}$ (ρ<2 kPa); $0.11\pm 0.07{\;\mathrm{kPa}}^{-1}$ (2<ρ<10 kPa)	millisecond	Bending radii down to 27 mm, sensitivity remains 80%	N/A	Cardiology, arterial stiffness	[64]
	$0.51{\;\mathrm{kPa}}^{-1}$	0.15 s	>40 compression (10,000 cycles)	N/A	Cardiology, early-stage Parkinson’s detection, muscle movement, vocalization waves	[65]
	$2254{\;\mathrm{kHz}\;\mathrm{mmHg}}^{-1}$	90 ms	N/A	Nearfield electromagnetic coupling	Cardiology, intracranial pressure	[66]
Piezoelectricity	10 kPa	~0.1 s	N/A	3 V	Cardiology, arterial stiffness	[67]
	$3.95{\;V\;N}^{-1}$	39 ms (static state)/27 ms (excited state)	Bending angle up to 90°	N/A	Cardiology, voice recognition	[70]
	~0.005 Pa	~0.1 ms	stretching ~30% with effective modulus of ~60 kPa	1–3 V	Cardiology, arterial stiffness	[68]
	(30 <ρ< 1800 kPa)	N/A	Stretching 30% (failure)	2–5 V	Dermatology	[69]

**Table 5 sensors-23-03673-t005:** Summary of useful adhesion duration.

Substrate Material	Device Thickness	Placement Site (Signal)	Exercise/Motion	Adhesion Length	On-Skin Measurement Duration	Ref
00-30 Ecoflex	37 μm	Skin (temperature), forehead (EEG), chest (ECG), leg (EMG)	Walking, skin stretching	N/A	6 h	[36]
Ecoflex 00-30	N/A	Wrist (blood pulse)	N/A	N/A	14 s	[68]
Ecoflex 00-30	20 μm	Skin (dermatology)	N/A	N/A	N/A	[69]
Ecoflex 00-30/ Bluestar 4642	0.9 mm	Arm (EMG), scalp (EEG)	Exercise, sleeping and showering, skin stretching	Consistent over several days	30 s	[50]
Ecoflex	~140 μm	Forearm, fingertip (blood flow)	Device compressing	N/A	30 min	[56]
PDMS	2 mm	Arm (ECG), scalp (EEG)	Swimming	N/A	70 s	[5]
PDMS	20 μm	Arm (ECG)	Wrist curl, squat, writing	N/A	40 s	[30]
PDMS	300 μm	Wrist (blood pulse)	N/A	N/A	10 s	[60]
PDMS	20 μm	Wrist (blood pulse), foot (pressure)	Walking, running, jumping	N/A	30 s	[61]
PDMS	N/A	Wrist (blood pulse)	Bike riding	N/A	1400 s	[63]
PDMS	700 μm	Skin (sweat)	Bike riding	N/A	1–6 h (one-time usage)	[228]
Adhesive PDMS	120 μm	Chest (ECG)	N/A	N/A	20 s	[49]
Adhesive Bluestar Silicones	2 mm	Chest (ECG), arm (EMG), neck (EEG)	N/A	N/A	20 s	[51]
SBS	0.1 mm	Wrist (blood pulse)	N/A	N/A	60 s	[66]
Silicone elastomer	3 μm	Ear (EEG)	Washing in soap water	2 weeks	3.5 min	[48]
PHB/PHV	2.4 mm	Wrist (blood pulse, PWV)	N/A	N/A	15 s	[64]
Silicone elastomer/PVA	~50 μm	Palm (temperature)	Device compressing	N/A	3 h	[4]
PVA	~30 μm	Bicep and tricep, lower back (EEG)	Skin stretching	2 weeks	30 s	[37]
PET	~30 μm	Chest (ECG, temperature)	Running	30 h	2000 s	[46]
PET	N/A	Skin (sweat)	Bike riding	N/A	7000 s	[221]
PEN	50 μm	Wrist (blood pulse, PWV)	N/A	N/A	3 s	[67]
Parylene	30 μm	$\mathrm{Fingertip}\;(\mathrm{PPG}/{\mathrm{SO}}_{2}$)	N/A	5 days	3 s	[59]
Polyurethane sponge	N/A	Wrist (blood pulse), static tremor (Parkinson’s disease diagnosis)	N/A	N/A	80 s	[62]

## Data Availability

No new data were created.

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
