# Peer review of "A Review of Skin-Wearable Sensors for Non-Invasive Health Monitoring Applications"

_sensors, 2023, doi:10.3390/s23073673_

Round 1

Reviewer 1 Report

pls see in attachment my comments and missed references

Author Response

We greatly appreciate the invaluable comments from our reviewers. We have revise our manuscript. The attached document highlighted a point-to-point response to all review comments. Thank you for your time!

Best,

Zhibin Yu

Associate Professor

Florida State University

Reviewer 2 Report

In this review, the authors comprehensively summarized the research progress of skin wearable sensors for non-invasive health monitoring. However, there are still several minor issues to be addressed:

1. It is suggested to use one three-line table in Table 1.

2. In the Introduction, you need to introduce the importance of your review and the differences between your and other published reviews.

3. There are some grammar and format mistakes. Please check the whole manuscript carefully.

Author Response

(The authors gave the same response as above.)

Reviewer 3 Report

The present manuscript performs a review on recent advancements in skin based wearable sensor for non-invasive disease diagnostic. Authors reported different sensing methods taking into account variations in different physiochemical parameters of skin, such as thermal, electrical, radiant, mechanical and electrochemical nature in the biofluid released from skin to predict the occurrence of an illness. The focus of these reviewed literatures is on the flexible and stretchable sensing devices reported recently for real application. I find the review manuscript interesting and an important contribution in the research area of sensors based non-invasive disease diagnostics. Thus, I recommend a minor revision and my comments are appended below.

1.      Authors highlighted different sensing methods adopted for skin based disease diagnostic. However, I believe that another key advancement in this area is the skin volatolomics. The detection of the release of VOCs from skin can serve as a basis of biomarkers based disease diagnostic. Some of the recent articles are (Doi: 10.3390/metabo12090824 ; 10.1038/srep04611 ; 10.1007/s12274-022-4459-3)

2.      Some grammatical errors and typos should be fixed. In page 2 “expended” and “costliness”. Title of 3.1 as “mechism”

Author Response

(The authors gave the same response as above.)
